# Multiwater Index Synergistic Monitoring of Typical Wetland Water Bodies in the Arid Regions of West-Central Ningxia over 30 Years

**Haiwei Pang** [1] 🄳**, Xinwei Wang** [2]**, Ruiping Hou** [3]**, Wanxue You** [2]**, Zhen Bian** [1,]*** and Guoqing Sang** [1]

[1]  The College of Water Conservancy and Environment, Jinan University, Jinan 250024, China
[2]  Administration Bureau of ShaPoTou National Nature Reserve, Zhongwei 751100, China
[3]  Academy of Forestry Inventory and Planning (AFIP), National Forestry and Grassland Administration, Beijing 100174, China
*   Correspondence: stu_bianz@ujn.edu.cn

**Abstract:** The Shapotou National Nature Reserve in the Ningxia Hui Autonomous Region is a typical arid region in China. There is an exceptionally serious problem of surface water resource conservation, and dynamic monitoring of surface water with the help of water indices can help to elucidate its change patterns and impact mechanisms. Here, we analysed the characteristics of interannual variation in surface water area in the study area from 1992–2021. The correlation coefficients of the surface water area in the previous year and the contemporaneous water bodies of the Yellow River with the total surface water area (TSWA) were calculated. The results show the following: ① In terms of the classification accuracy of the two methods, water indices and support vector machine classification, water indices are more suitable for water body extraction in the study area. In particular, the three water indices, NDWI, MNDWI and AWEIsh, were more effective, with average overall accuracies of 90.38%, 90.33% and 90.36% over the 30-year period, respectively. ② From the TSWA extraction results from the last 30 years, the TSWA showed an increasing trend with an increase of 368.28 hm$^2$. Among the areas, Tenggeli Lake contributed the most to the increase in TSWA. ③ The highest correlation between the TSWA and the previous year's TSWA was 0.89, indicating that the better way to protect the water body is to maintain water surface stability year-round. The surface water area of the Yellow River and TSWA also showed a strong correlation, indicating that the rational use of Yellow River water is also an important direction for the future conservation of water resources in the study area.

**Keywords:** water index; surface water area; remote sensing; interannual variation; trend variation; correlation

## 1. Introduction

In arid and semiarid regions, water resources play a very important role in maintaining surface vegetation and stabilising ecosystems [1–3]. With the dual effects of global warming and human activities, the arid and semiarid regions of northwest China are gradually showing warming–drying climate trends, and the importance of water resources has become more significant [4,5]. In this context, the comprehensive development, utilisation and protection of water resources in this region is particularly important. Quickly and accurately extracting the surface water area is a prerequisite for monitoring watershed changes [6–8].

Remote sensing technology has become a common tool for water resource monitoring due to its advantages, such as a large sensing range and high timeliness [9,10]. Based on the spectral reflectance difference between water bodies and other features, researchers have constructed various water indices for the automatic extraction of water bodies [6,11,12]. McFeeters (1996) first proposed the normalised difference water index (NDWI), which uses

the green and near-infrared bands of the Landsat thermal mapper (TM). This index can weaken the influence of surface cover conditions, such as soil vegetation, while enhancing the spectral characteristics of water to more effectively extract water bodies such as lakes and reservoirs [13]. Nearly a decade later, Xu (2005) built the modified normalised difference water index (MNDWI) based on the NDWI, improving on the NDWI by extracting town-wide water bodies and easily distinguishing between shadows and water bodies [14]. Since 2005, new water indices have been proposed for almost every year. Regarding surface water extraction in semiarid areas of China, Yan (2007) proposed the enhanced water index (EWI), which effectively distinguishes semidry rivers from background noise in semiarid regions [15]. This is the first time that researchers have constructed a water index for water bodies in China's semiarid regions. Ding (2009) proposed the new water index (NWI) based on the strong absorption abilities of water bodies in the NIR and mid-infrared bands and verified that the NWI has strong generalisability to different water body types [16]. Feyisa (2014) proposed a new automatic water extraction index (AWEI) and further developed it into AWEIsh and AWEInsh by including and omitting a shadow-removal capability to overcome the effects of low-reflectance image features on the extraction effect; these indices have been used to extract water bodies from Landsat images [17]. AWEI's two versions of the water index explain some of the shadow and dark surface problems that the water index has difficulty addressing [18]. Xu (2021) surmised that the most widely used indices over the last 20 years have been the MNDWI and NDWI, while the AWEIsh ranks third, and a review of water indices has been produced over the past two decades. [19]. The key to water feature extraction based on the water index is threshold selection is to distinguish water and non-water regions [20]. At present, the commonly used threshold selection methods include manual selection and image threshold algorithms [21–23]. However, with the deepening of the research, it is difficult for the local optimal threshold to be universal. Due to the influence of unavoidable factors such as atmospheric conditions and aerosols, the threshold values of different time ranges and different types of water bodies are different. Therefore, determining the threshold value has become the key point for the application of the water index [24].

Since its launch in 1972, data have been obtained from Landsat satellites, where the spatial resolution in the visible to shortwave infrared bands has undergone resolution improvements from 80 m to 30 m, starting with the early multispectral scanner (MSS) and evolving into the TM onboard Landsat 4/5, the Enhanced Thematic Mapper Plus (ETM+) onboard Landsat-7 and the Operable Land Imager (OLI) onboard Landsat-8 [25–28]. Landsat-9 satellite with Operational Land Imager 2 (OLI-2) launched in 2021. The surface water body results extracted using these data are very satisfactory [29]. For example, DU et al. extracted the water body distributions in various regions of the Yangtze River basin and Huaihe River basin in China from Landsat OLI images [30]. Rokni used LandsatTM, ETM+ and OLI images to extract the area of Lake Urmia from 2000 to 2013 and explored its spatial and temporal variations. G-EAU et al. used Landsat 5, 7 and 8 images to monitor the areas of Xiaohu Lakes and small reservoirs [31]. Landsat imagery has been commonly used in surface water monitoring research because these moderate resolution images are continuously provided free of charge [32,33].

It is generally believed that different water indices have different extraction effects on different types of water. Therefore, different water indices for water body information extraction were chosen for different types of water bodies in the study area. Five remote sensing water indices, NDWI, MNDWI, AWEIsh, EWI and NWI, were selected preliminarily. At the same time, the accuracy sampling points were selected based on Google Earth satellite images, and the confusion matrix was calculated to verify the extraction accuracy. According to the results of visual extraction and confusion matrix verification, the best water indices of TSWA, lake, reservoir and Yellow River extraction in the study area were determined. The support vector machine (SVM) classification method was added to further verify the reliability of the water index and avoid the subjectivity of human visual observation. In this paper, LandsatTM, ETM+ and OLI series data collected in the past

30 years (1992–2021) were used as remote sensing data sources to extract water bodies in the study area. The work has three main objectives. The first objective is to obtain a 30-year time series surface water area dataset through the water index. The second objective is to analyse the characteristics of surface water changes over many years. The third objective is to analyse the correlation between TSWA change in the study area and the surface water area of the Yellow River and the previous year TSWA to study its influencing mechanism. The results of this study can help support researchers' understanding of the long-term and dynamic surface water changes occurring in the study area and have important practical significance for the scientific management of water resources in the study area.

## 2. Materials and Methods

To achieve the research objectives of this paper, the main operation steps are as follows: collection of the data from the study area from 1992 to 2021 (mainly including Landsat data, meteorological data and Google Earth satellite images); image pre-processing of Landsat series data (radiometric calibration, atmospheric correction). The water index layer is calculated. The optimal water index is determined based on the confusion matrix, and the surface water area during the 30-year period is extracted in order to analyse the variation characteristics and influencing mechanism of surface water. To better explain the surface water extraction method, a working flow chart is shown in Figure 1.

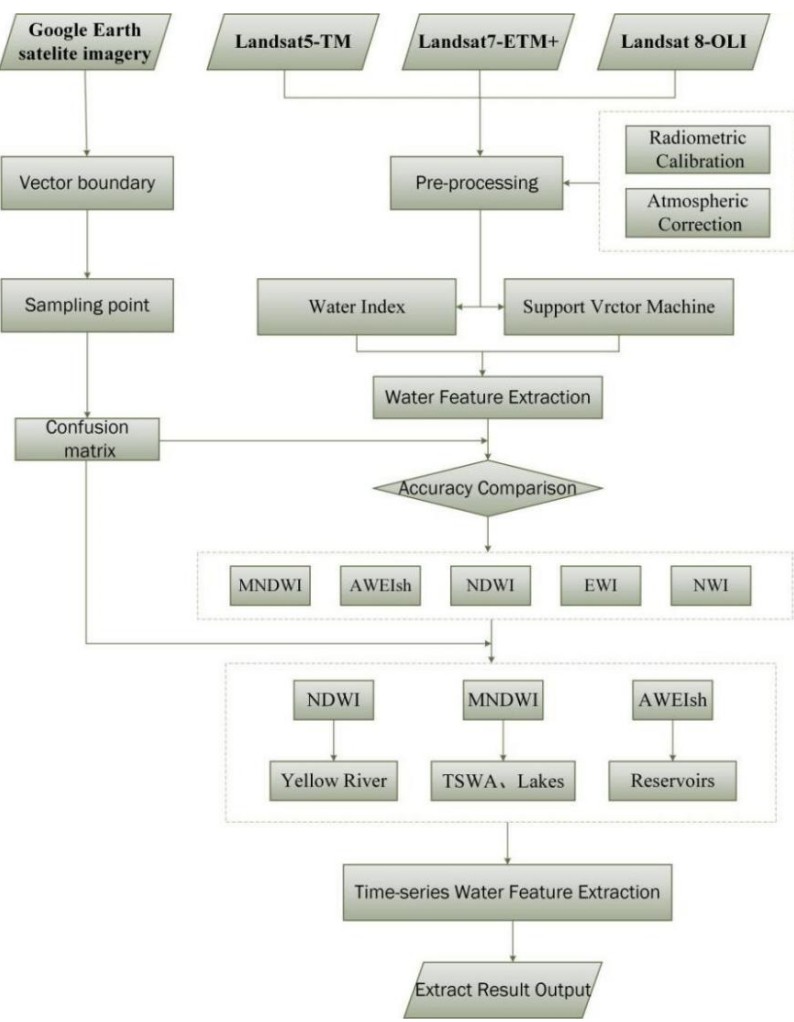

**Figure 1.** Workflow diagram.

*2.1. Study Area*

The study area considered herein is located in Shapotou National Nature Reserve, Zhongwei city, Ningxia Hui Autonomous Region, China, with a total area of 273.49 km$^2$. This region is located at the south-eastern edge of the Tengger Desert, with geographical coordinates of 104°49′25″~105°09′24″ E and 37°25′58″~37°37′24″ N. It has a typical continental arid climate, with an average annual temperature of 9.13 °C (1959–2021, the same as below) and an average annual precipitation total of 185.94 mm. The average annual evaporation over the past 10 years was 2051.52 mm (2008–2017), approximately 10 times the average annual precipitation and precipitation has been concentrated from May–August; evaporation greater than 150 mm occurs in all months. The average temperature in the last ten years was 10.39 °C (2012–2021, the same below), and the average annual precipitation was 215.02 mm; this precipitation was low and mostly concentrated in summer. Since the reserve is located on the northern edge of the East Asian monsoon region, the climate is highly susceptible to changes in the strength of the East Asian monsoon, thus resulting in large interannual temperature and, especially, precipitation variations. In addition, the variabilities in this climate system lead to the increased vulnerability of the regional ecological environment. (Precipitation data were obtained after calculation. The evaporation data were referenced from the research results of the Ningxia Zhongwei Shapotou National Nature Reserve Administration and the Institute of Cold and Arid Zone Environment and Engineering of the Chinese Academy of Sciences. In the "Ningxia Zhongwei Shapotou National Nature Reserve-Phase III Comprehensive Scientific Investigation Report" prepared in 2021).

The water resources in the study area mainly include atmospheric precipitation, surface runoff, groundwater and water stored in lakes, ponds and reservoirs. There are few perennial rivers in the area; only in the western part of the protected Chang Liushui area do springs outcrop to form rivers, and these rivers are intercepted by reservoirs and dams; in addition, there are many breaks in the Chang Liushui area. There are many lakes and ponds in the protected area, and the largest lakes include Tenggeli Lake, Xiaohu Lake, Gaodun Lake, Qiandao Lake and Machang Lake. The reservoirs include Mengjiawan Reservoir and Changliushui Reservoir. Considering the influence of groundwater volume interactions on the water area in this protected area, it is necessary to analyse the concurrent changes in the water area of the Yellow River. The Yellow River flows through Zhongwei City, with a length of 182.4 km in Zhongwei City. It is close to Shapotou National Nature Reserve. In order to present a complete picture of the location of the Yellow River in relation to the study area, we have extended the display area to include all the towns in the city of Zhongwei through which the Yellow River flows, so the area shown in Figure 1 includes the whole area of the Yellow River flowing through Zhongwei city (Figure 2).

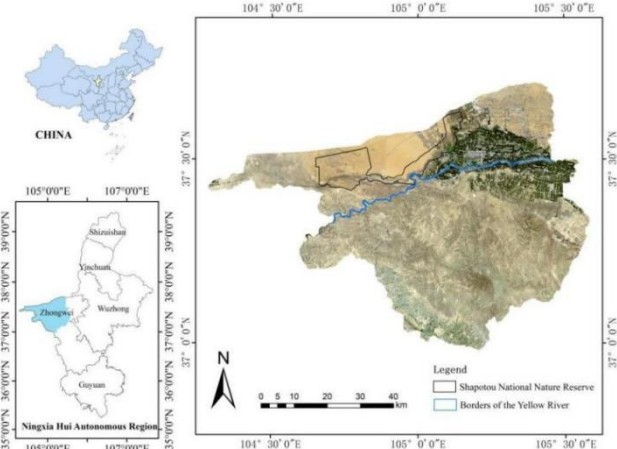

**Figure 2.** Location of the study area (the right-hand side of the image shows Landsat-7 ETM+ data from March 2019).

*2.2. Data*

2.2.1. Landsat Series Data

This paper uses the Landsat Collection2 Level-1 dataset with a time series of 1992–2021, with one period selected each year (March data), with April and May data substituted for individual years in which March images are missing (Table 1). The main reasons for selecting data from this period are as follows: during this period, the temperature in the study area warmed, frozen waters thawed and the waters were in a more abundant state, which facilitated the detection of long-time sequences of waters, while the study by Chen et al. [34] pointed out that spring represents the lowest cloud period of the year in the Ningxia region, which mitigates the effects of clouds on water extraction. (Data for 1993 and 1996 are missing from this paper).

All images were obtained from the United States Geological Survey (USGS) website. To control the influence of cloudiness on the water extraction accuracy, all images were controlled to below 5% cloudiness and pre-processed by radiometric calibration and atmospheric correction, thus ensuring good data quality [35,36].

**Table 1.** Data used in the present study.

| Type of Data | Source | Date |
|---|---|---|
| Landsat-5 TM | USGS | 1992/04/01; 1994/03/06; 1995/03/25; 1997/03/30; 1998/04/02; 1999/04/05; 2002/03/28; 2005/03/04; 2007/05/13; 2008/04/13; 2009/03/15; 2010/04/03 |
| Landsat-7 ETM+ | USGS | 2000/03/30; 2001/03/17; 2003/03/23; 2004/03/25; 2006/03/15; 2011/03/29; 2012/03/31; 2013/04/03; 2014/03/21; 2015/03/24; 2016/03/26; 2019/03/03; 2020/03/21; 2021/02/20; 2021/03/08; 2021/05/11; 2021/11/03 |
| Landsat-8 OLI | USGS | 2017/04/22; 2018/03/08; 2021/01/11; 2021/06/04; 2021/07/22; 2021/08/07; 2021/09/08; 2021/11/03; 2021/12/13 |

Landsat-7 ETM+ data are missing after 2003 due to a malfunction of the ETM scanline corrector; thus, the problem of missing strips occurs, but the data still preserve good radiometric and geometric properties [37]. In this paper, we use ENVI 5.3 software to implement the corresponding gap-fill correction, as shown in Figure 3.

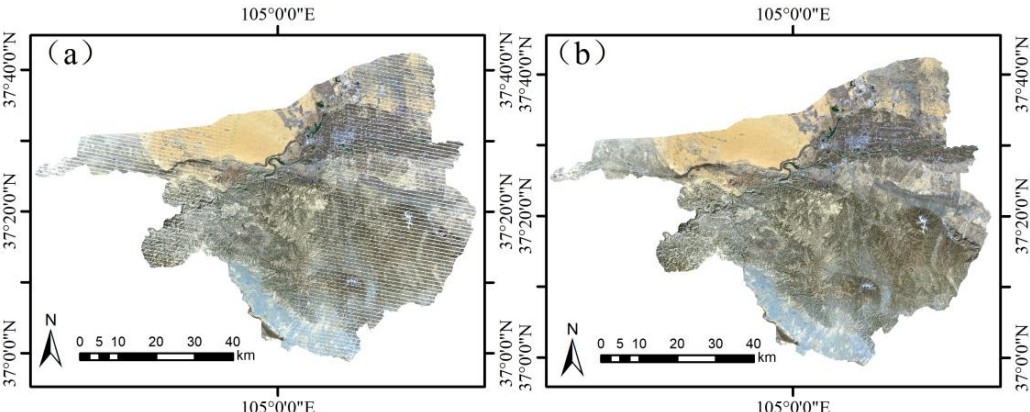

**Figure 3.** Repair of missing strip problems: (**a**) before restoration, (**b**) after the repair was completed, the data shown are Landsat -7 ETM+ and were collected on 3 March 2019.

2.2.2. Google Earth Satellite Imagery

Google Earth satellite imagery is used in this paper, mainly due to its higher spatial resolution of 0.5 m covering the entire watershed of the reserve. Since Google Earth images are stitched together from multisensor data, large-area images are not temporally consistent [38]. The image depicting the eastern main lake waters used in this paper was taken on 21 March 2019, and the image acquisition time for both Mengjiawan Reservoir and Changliushui Reservoir in the western study area was 12 July 2018. These images are shown in Figure 4. The remaining years of Google Earth satellite imagery are obtained from Google Earth software and can be directly mapped in the software surface boundary.

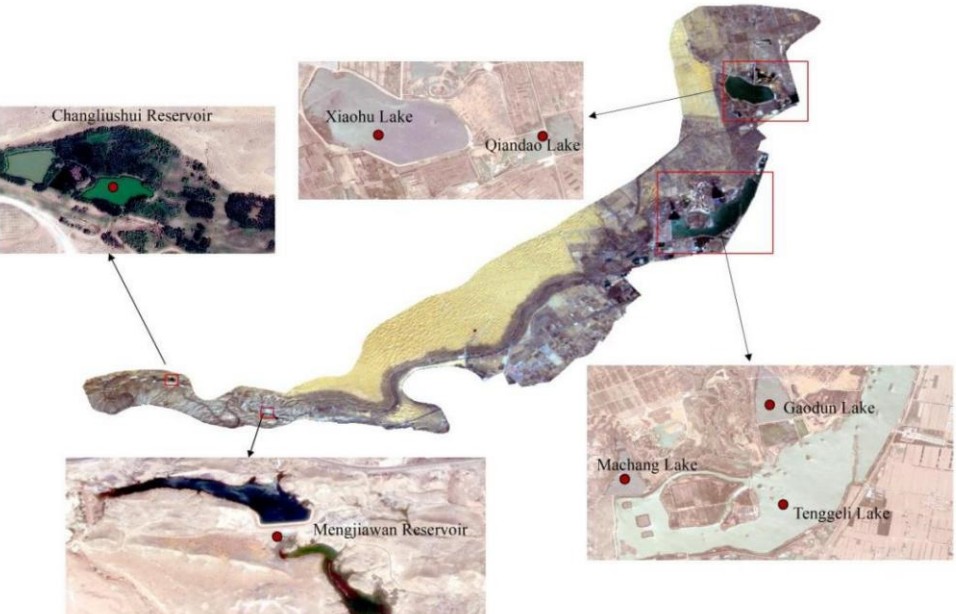

**Figure 4.** Google Earth satellite imagery of water bodies. To present the best display image, some areas without water bodies are not shown. Each individual area shown is a Google Earth satellite imagery. The complete main image in the middle is the March 2019 Landsat-7 data.

*2.3. Methods*

2.3.1. Water Indices

From the above review of water body indices developed over the past two decades, it can be found that NDWI, MNDWI and AWEIsh are the most widely used water indices that have undergone experiments by a large number of researchers, verifying their efficiency in water body extraction with sufficient guarantees of reliability. However, the creators of EWI point out in their study that EWI can distinguish water bodies from background noise in arid regions. Our study area is arid; similarly, for NWI, its creators note that it can effectively distinguish between various types of water bodies and the study area of this paper contains a variety of water bodies, such as lakes and reservoirs. Finally, we decided to use NDWI, MNDWI, EWI, AWEIsh and NWI for the initial extraction of water bodies in the study area, mainly including Tenggeli Lake, Xiaohu Lake, Gaodun Lake, Machang Lake, Mengjiawan Reservoir and the Yellow River (flowing through the study area). The calculation formula of each water body index is as follows [13–17]:

$$NDWI = \frac{\rho GREEN - \rho NIR}{\rho GREEN + \rho NIR} \tag{1}$$

$$MNDWI = \frac{\rho GREEN - \rho SWIR}{\rho GREEN + \rho SWIR} \tag{2}$$

$$EWI = \frac{\rho GREEN - \rho NIR - \rho MIR}{\rho GREEN + \rho NIR + \rho MIR} \tag{3}$$

$$AWEIsh = \rho BLUE + 2.5\rho GREEN - 1.5(\rho NIR + \rho SWIR1) - 0.25\rho SWIR2 \tag{4}$$

$$NWI = \frac{\rho BLUE - \rho NIR - \rho SWIR1 - \rho SWIR2}{\rho BLUE + \rho NIR + \rho SWIR1 + \rho SWIR2} \tag{5}$$

where *NDWI* is the normalised differential water index; *MNDWI* is the modified normalised differential water index; *EWI* is the enhanced water index; *AWEI*sh is the automatic water body extraction index; *NWI* is the new water index; $\rho_{BLUE}$ is the surface reflectance in the blue band; $\rho_{GREEN}$ is the surface reflectance in the green band; $\rho_{NIR}$ is the surface reflectance in the NIR band; $\rho_{MIR}$ is the surface reflectance in the SWIR band; $\rho_{SWIR1}$ is the surface reflectance in SWIR band 1; and $\rho_{SWIR2}$ is the surface reflectance in SWIR band 2.

2.3.2. Accuracy Verification

To verify the accuracy of the water extraction results, this paper determines the final extraction effect by constructing a confusion matrix [39]. The calculation of the confusion matrix is carried out with the help of sampling points. Sampling points were obtained from higher resolution images. In high-resolution satellite images, it is relatively easy to visually distinguish between surface and non-surface, so it is reliable to verify the accuracy. Confusion matrices were calculated for each extraction result using ENVI software and sampling point data to obtain the overall accuracy and Kappa coefficients, which were used to demonstrate the accuracy of the results.

The selection of sampling points in this paper is based on the Google Earth satellite image with a resolution of 0.5 m. Except for Google Earth satellite imagery data purchased manually in 2019 (TIFF format), Google Earth satellite imagery was obtained from Google Earth software and the surface boundary was directly depicted in the software. First, based on Google Earth satellite images, the surface boundary is visually depicted and the surface boundary vector layer is generated. Then, the random point creation tool in ArcGIS 10.7 software was used to randomly generate sampling points on the vector layer of the water surface.

To ensure the accuracy of accuracy verification, through the comparison of Landsat image data over the years, it was found that there were three major periods of water body area change in the study area, namely, 1992–2003, 2004–2009 and 2010–2021. Since the overall water area did not change in a large area in each period, three sets of precision sampling points were constructed in this paper. In the year of the construction of sampling points, the year with the largest water area in each period is judged according to the vision to ensure the maximum number of sampling points to avoid the phenomenon of excessive accuracy caused by the small number of sampling points. The specific information of the sampling points is shown in Table 2. The sampling points in 2019 are mapped (Figure 5), and the main water bodies are enlarged and displayed.

**Table 2.** Number of sampling points.

| Time Range | Sampling Point Year | Tenggeli Lake | Xiaohu Lake | Gaodun Lake | Machang Lake | Qiandao Lake | Mengjiawan Reservoir | Changliushui Reservoir | Yellow River |
|---|---|---|---|---|---|---|---|---|---|
| 1992–2003 | 2003 | 100 | 75 | 100 | 60 | 0 | 20 | 5 | 104 |
| 2004–2009 | 2009 | 100 | 100 | 100 | 52 | 100 | 22 | 4 | 102 |
| 2010–2021 | 2019 | 100 | 100 | 100 | 47 | 74 | 18 | 4 | 102 |

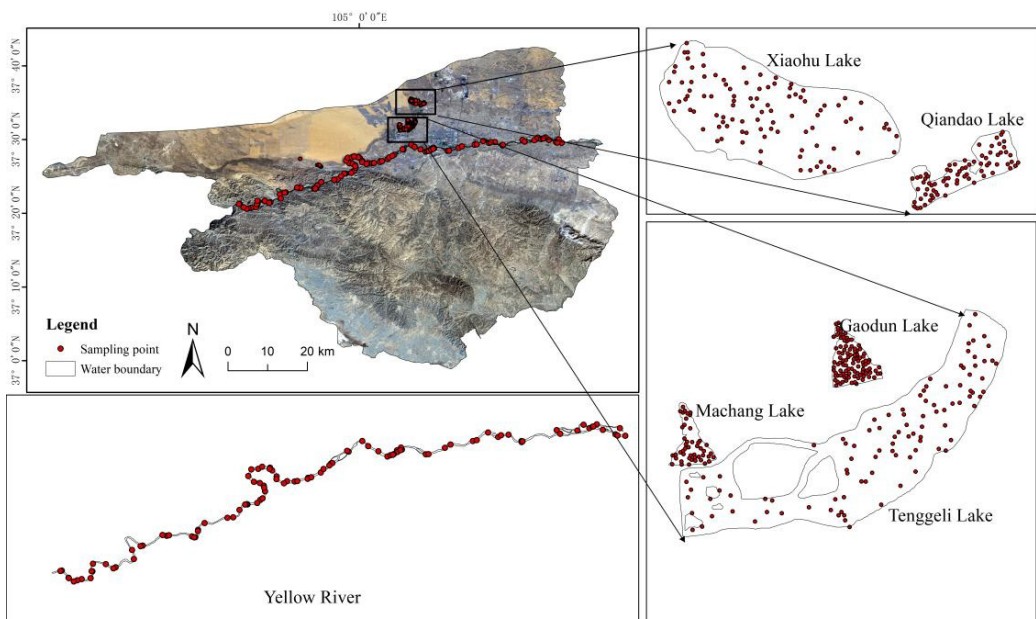

**Figure 5.** Sample point example (The sample points are shown in the figure).

### 2.3.3. Threshold Selection

The selection of the threshold is related to the accuracy of the final water extraction. In this paper, a confusion matrix is used to determine the optimal threshold for the water index in every year. Usually, 0 is set as the initial threshold, but every time the threshold starts from 0, it needs to invest a lot of time. Based on the data in 2019, this paper conducted the first threshold experiment, taking 0.05 and 500 as step sizes to determine the optimal thresholds of each water index in this year. Then, based on the results of the threshold experiment in 2019, the optimal threshold was set as the initial threshold of the threshold experiment in other years due to the similar spectral characteristics of interannual water bodies, thus saving the time spent for each threshold starting from 0. According to the above experimental design, the optimal threshold of the time series for 30 years was determined and the water body was extracted to obtain the time series dataset of surface water in the study area.

### 2.3.4. Support Vector Machine Classification

Support vector machines not only satisfy the classification requirements by establishing the optimal classification hyperplane but also ensure that the classification accuracy is maximised [40]. At the same time, the SVM uses the training error as a constraint on the optimisation problem and minimises the confidence interval as the optimisation objective, with the solution being the only optimal solution [41]. Unlike the extraction of water bodies with the help of water indices, SVM does not require the determination of thresholds but only the selection of training samples. Since we have already determined the locations of various water bodies in the study area and can visually determine the water body areas with the help of Google Earth satellite imagery and Landsat data, the accuracy of sample selection is guaranteed, which is why we have chosen the support vector machine method. Another purpose of choosing the support vector machine classification method is to compare it with the water index method. The final results of the SVM can verify the feasibility of the water index method.

The support vector machine classification method was manipulated through the ENVI software. The final accuracy validation method still uses the confusion matrix to calculate the overall accuracy and the determination of the Kappa coefficients.

### 2.3.5. Trend Slope Analysis Method

The slope of the interannual rate of change in the water body area time-series in the study area is calculated as follows [34]:

$$Slope = \frac{n \times \sum\limits_{i=1}^{n}(i \times S_i) - \sum\limits_{i=1}^{n} i \sum\limits_{i=1}^{n} S_i}{n \times \sum\limits_{i=1}^{n} i^2 - \left(\sum\limits_{i=1}^{n} i\right)^2} \tag{6}$$

where $S_i$ is the water body area in year $i$ and $n$ is the total number of years; when *slope* > 0, the water body area exhibits an increasing trend during the study period and when *slope* < 0, the water body area exhibits a decreasing trend during the study period.

The Mann–Kendall method is used to test the significance of the areal change in each water body type during the study period; this method does not require the samples to obey a certain regular distribution and can prevent interference resulting from a few outliers. It is a nonparametric statistical test method that has been widely used in climate ecology research and other related fields [42,43]. The calculation formulas are as follows:

$$S = \sum\limits_{k=1}^{n-1} \sum\limits_{j=k+1}^{n} Sgn(x_j - x_k) \tag{7}$$

$$Sgn(x_j - x_k) = \begin{cases} 1 & (x_j - x_k) > 0 \\ 0 & (x_j - x_k) = 0 \\ -1 & (x_j - x_k) < 0 \end{cases} \tag{8}$$

where $S$ is a normal distribution function with a mean of 0 and a variance of $var(S) = n(n-1)(2n+5)/18$. When $S > 0$, the standard normal statistic variable is calculated using the following equation:

$$Z = \begin{cases} \dfrac{S-1}{\sqrt{var(S)}} & (S > 0) \\ 0 & (S = 0) \\ \dfrac{S+1}{\sqrt{var(S)}} & (S < 0) \end{cases} \tag{9}$$

where $Z$ is a statistical value; the trend is increasing when $Z$ is greater than 0 and decreasing when $Z$ is less than 0. Absolute values of $Z$ greater than 1.645, 1.960 and 2.576 indicate that the significance test is passed at the 90%, 95% and 99% confidence levels, respectively.

### 2.3.6. Pearson Correlation Coefficient

Pearson correlation coefficients were analysed to assess the correlation degrees between the TSWA and precipitation, temperature, the previous-year TSWA and the contemporaneously extracted Yellow River area results using the following equation:

$$r = \frac{\sum_{i=1}^{n}(x_i - \overline{x})(y_i - \overline{y})}{\sqrt{\sum_{i=1}^{n}(x_i - \overline{x})^2} \times \sqrt{\sum_{i=1}^{n}(y_i - \overline{y})^2}} \tag{10}$$

where $r$ is the Pearson correlation coefficient, $n$ is the total number of years and $x$ and $y$ are the variables sought. When $|r| \geq 0.7$, the two variables are considered to be strongly correlated; when $0.4 \leq |r| < 0.7$, they are considered to be moderately correlated; when $0.2 \leq |r| < 0.4$, they are weakly correlated; and when $|r| \leq 0.2$, the two variables are considered to be extremely weakly correlated or not correlated. The same

significance test was carried out at the 95% and 99% significance levels ($p < 0.05$ and $p < 0.01$, respectively) [44,45].

## 3. Results and Discussion

### 3.1. Initial Experimental Threshold Selection Results

Based on the Landsat data from March 2019, the threshold experiments of each water index were conducted. For the first time, 0 was taken as the threshold for each water index. Except for AWEIsh, which took 500 as the step size, all the other water indices took 0.05 as the step size. The extraction results were calculated using the sampling points to calculate the confusion matrix and the precision results are shown in Figure 6, indicating that with an increase in the threshold value, the extraction accuracies of all five indices showed trends of first increasing and then decreasing. The NDWI had the highest accuracy at the threshold value of −0.2, with an overall accuracy of 93.63% and a Kappa coefficient of 0.87. The MNDWI had the highest accuracy at the threshold value of −0.15, with an overall accuracy of 96.32% and a Kappa coefficient of 0.93. The EWI had the highest accuracy at a threshold value of −0.50, with an overall accuracy of 96.35% and a Kappa coefficient of 0.89. The AWEIsh had the highest accuracy at the threshold value of −2000, with an overall accuracy of 96.24% and a Kappa coefficient of 0.92. Finally, the NWI had the highest accuracy at the threshold value of −0.75, with an overall accuracy of 94.32% and a Kappa coefficient of 0.89. Compared with setting the initial experimental threshold as 0, the selection efficiency of the optimal threshold can be further improved through the experimental results of the threshold. In subsequent experiments, the initial experimental thresholds of NDWI, MNDWI, EWI, AWEIsh and NWI started from −0.2, −0.15, −0.50, −2000 and −0.75, respectively.

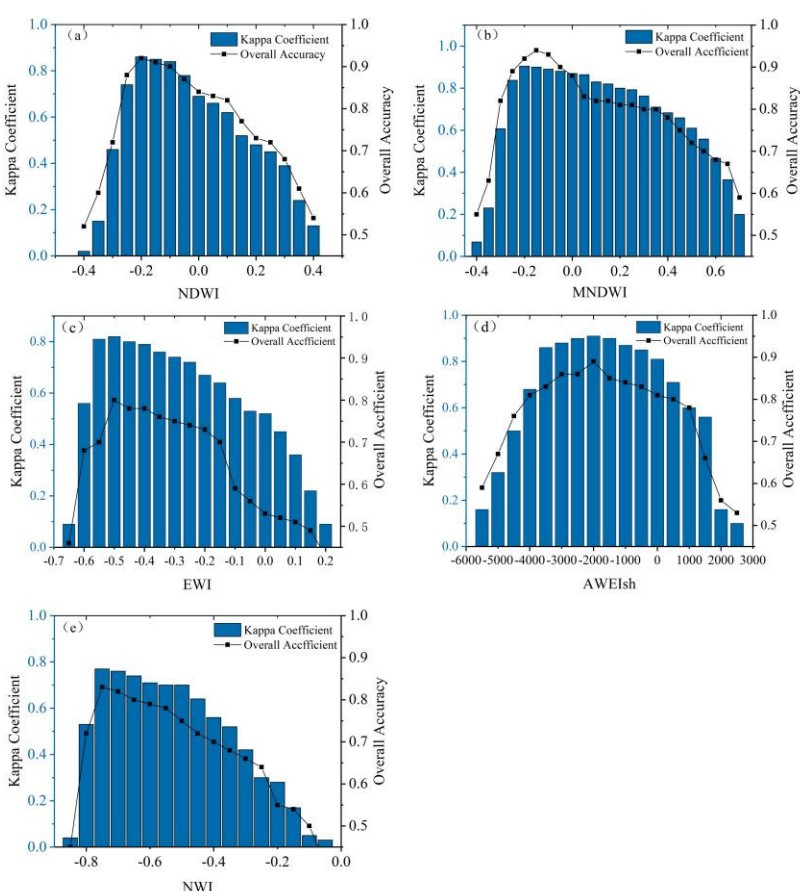

**Figure 6.** Accuracy of threshold experiments: (**a**) NDWI threshold test, (**b**) MNDWI threshold test, (**c**) EWI threshold test, (**d**) AWEIsh threshold test, (**e**) NWI threshold test.

*3.2. Water Index and SVM Water Extraction Accuracy*

Using the water index and Landsat data from the last ten years, the water index threshold method and SVM method were used to extract the TSWA in the study area. The reliability of the water index is verified by comparing the overall accuracy of the two methods by calculating the confusion matrix combined with the total sampling points.

3.2.1. Extraction Accuracy of Water Index

Based on the initial threshold of each water index determined in Section 3.1, the TSWA was extracted from the study area during 2012–2021. Except for the AWEIsh step size of 500, the rest of the water index step sizes are still 0.05. By calculating the confusion matrix, the optimal threshold value is determined when the precision value is the highest and surface water extraction is completed, as shown in the Table 3. Among them, the average accuracies of NDWI, MNDWI, EWI, AWEIsh and NWI in the last ten years were 91.14% (kappa = 0.84), 91.33% (kappa = 0.85), 81.03% (kappa = 0.71), 90.05% (kappa = 0.84) and 81.15% (kappa = 0.75), respectively. Among them, EWI has the lowest accuracy. The analysis of the reasons shows that in the construction process of EWI, the near-infrared (NIR) and mid-infrared (MIR) bands are combined together, and the ratio calculation is carried out with the green light band. As the concentration of suspended solids in water increases, the reflectance of NIR and MIR to water increases, so it is no longer applicable to turbid water. As the study area is mostly lakes and reservoirs, the relatively closed environment makes the suspended algae easy to develop in large quantities, resulting in low extraction accuracy. Although the NWI index is pointed out by its builder as having strong applicability to different types of water bodies, the extraction effect in this study area obviously does not meet the extraction requirements through the experiment in this paper.

**Table 3.** Extraction accuracy of the water index.

| Date | NDWI | | | MNDWI | | | EWI | | |
|---|---|---|---|---|---|---|---|---|---|
| | Threshold | OA | Kappa | Threshold | OA | Kappa | Threshold | OA | Kappa |
| 2012/3/31 | −0.15 | 90.16% | 0.83 | −0.15 | 90.42% | 0.84 | −0.5 | 79.15% | 0.68 |
| 2013/4/3 | −0.15 | 90.89% | 0.83 | −0.1 | 92.15% | 0.87 | −0.45 | 80.73% | 0.74 |
| 2014/3/21 | −0.15 | 93.37% | 0.85 | −0.15 | 89.34% | 0.83 | −0.4 | 83.65% | 0.75 |
| 2015/3/24 | −0.15 | 92.51% | 0.86 | −0.1 | 93.68% | 0.89 | −0.5 | 82.96% | 0.77 |
| 2016/3/26 | −0.15 | 89.65% | 0.80 | −0.1 | 89.56% | 0.87 | −0.5 | 80.18% | 0.64 |
| 2017/4/22 | −0.15 | 89.91% | 0.84 | 0 | 89.30% | 0.76 | −0.5 | 82.45% | 0.79 |
| 2018/3/8 | −0.15 | 90.22% | 0.81 | −0.1 | 89.36% | 0.85 | −0.45 | 84.71% | 0.78 |
| 2019/3/3 | −0.2 | 91.72% | 0.86 | −0.15 | 94.23% | 0.90 | −0.5 | 81.22% | 0.72 |
| 2020/3/21 | −0.2 | 90.78% | 0.88 | −0.1 | 93.75% | 0.88 | −0.5 | 73.87% | 0.54 |
| 2021/3/8 | −0.15 | 92.16% | 0.83 | −0.1 | 91.48% | 0.85 | −0.5 | 81.36% | 0.73 |
| | AWEIsh | | | NWI | | | | | |
| 2012/3/31 | −2000 | 88.10% | 0.8 | −0.75 | 81.38% | 0.72 | | | |
| 2013/4/3 | −2000 | 92.78% | 0.81 | −0.75 | 79.51% | 0.75 | | | |
| 2014/3/21 | −2000 | 91.15% | 0.90 | −0.6 | 84.02% | 0.75 | | | |
| 2015/3/24 | −2000 | 86.42% | 0.86 | −0.65 | 86.97% | 0.83 | | | |

**Table 3.** *Cont.*

| Date | NDWI | | | MNDWI | | | EWI | | |
|---|---|---|---|---|---|---|---|---|---|
| | Threshold | OA | Kappa | Threshold | OA | Kappa | Threshold | OA | Kappa |
| 2016/3/26 | −2000 | 88.28% | 0.81 | −0.7 | 73.42% | 0.66 | | | |
| 2017/4/22 | −2000 | 90.91% | 0.80 | −0.75 | 81.12% | 0.80 | | | |
| 2018/3/8 | −2000 | 89.77% | 0.82 | −0.75 | 81.36% | 0.76 | | | |
| 2019/3/3 | −2000 | 92.16% | 0.89 | −0.75 | 83.25% | 0.77 | | | |
| 2020/3/21 | −2000 | 90.66% | 0.86 | −0.7 | 88.85% | 0.85 | | | |
| 2021/3/8 | −1500 | 90.31% | 0.85 | −0.75 | 71.57% | 0.65 | | | |

### 3.2.2. SVM Extraction Accuracy

Support vector machine classification was performed using Landsat data for the last ten years from 2012–2021. In the classification process, the water body of interest (water samples) was selected with reference to Google Earth satellite images and the classification accuracy and Kappa coefficients are shown in Table 4. The same sampling points as the water index are used in the accuracy verification process. Comparing Table 3, it is easy to determine that the accuracy of the classification method using support vector machines is low, although in 2012, 2013 and 2014, the accuracy is high, but the performance of the Kappa coefficient is still not optimal. Although the SVM method avoids this stage of threshold selection, it is equally important for the selection of training samples, which is directly related to the level of classification accuracy. As seen in Table 4, the average overall accuracy was 88.46% over the last ten years. The SVM extraction effect in this paper is still lower than that of the water index method.

**Table 4.** Support vector machine classification accuracy (2012–2021).

| Date | OA | Kappa |
|---|---|---|
| 2012/03/31 | 91.11% | 0.68 |
| 2013/04/03 | 90.93% | 0.73 |
| 2014/03/21 | 91.75% | 0.79 |
| 2015/03/24 | 84.33% | 0.54 |
| 2016/03/26 | 84.56% | 0.54 |
| 2017/04/22 | 89.67% | 0.57 |
| 2018/03/08 | 87.58% | 0.48 |
| 2019/03/03 | 87.82% | 0.56 |
| 2020/03/21 | 87.44% | 0.62 |
| 2021/03/08 | 89.37% | 0.72 |

### 3.3. Water Index Accuracy of Different Types of Water Bodies

By comparing the SVM classification results, it has been determined that the water index in this study has a better extraction effect. For different types of water, each water index has a different extraction effect. To extract different types of water bodies (lake, reservoir and Yellow River) in the study area more accurately, the sampling points of the lake, reservoir and Yellow River are used to calculate the extraction accuracy of different types of water bodies in the last ten years and the group with the highest accuracy is tabulated, as shown in Table 4. It can be seen that for lakes, reservoirs and the Yellow River in the study area, the highest extraction accuracies are for the MNDWI, AWEIsh and NDWI,

and the average overall accuracies are 91.68%, 90.79% and 90.97, respectively. However, to determine the final water index, artificial visual judgement is needed to extract the effect.

### 3.4. Extraction Effect of Different Types of Water

For the visual extraction effect, the accurate extraction of small water bodies is an important basis for judgement. Based on the data in 2019, the extraction effects of each water index on different types of water bodies were analysed, and the applicable water index was finally determined.

#### 3.4.1. Total Surface Water Area (TSWA)

The optimal threshold of each water index was used to extract the TSWA of the study area. Since lakes occupy a large proportion of the total surface water area of the study area and there are many small water bodies in the Tengger Lake area, the TSWA will be more accurate if lakes and small water areas can be accurately extracted. Therefore, Tengger Lake, Gaodun Lake and Machang Lake are taken as examples to show the extraction effect of each water index, as shown in Figure 7. Compared to Figure 4, the NDWI-extracted water body areas are small compared to the other water body index extraction results; thus, the NDWI failed to accurately reflect the distribution of water bodies in the protected area, while the NWI-extracted water body area was relatively large and exhibited a partial misclassification phenomenon (the obvious hardening of a road was classified as a water body). The Tenggeli Lake, Gaodun Lake and Machang Lake areas are shown separately. The MNDWI-extracted results show good extraction effects for the fine water bodies around the lake, while the rest of the indices could not completely extract these water bodies. The MNDWI uses the SWIR band instead of the NIR band in the NDWI, which weakens the influence of soil and buildings, thus making the extraction results of fine water bodies in the Tenggeli Lake area more accurate; therefore, the MNDWI index was used to extract the total water body area in the protected area.

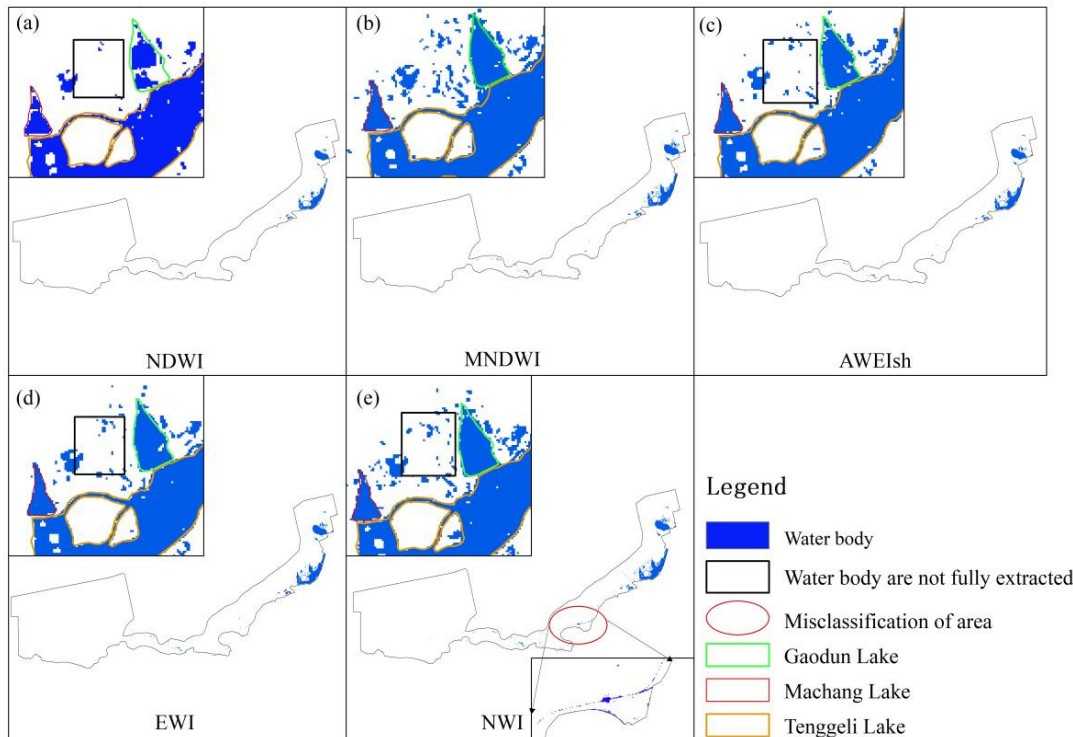

**Figure 7.** Extraction of images for effects. (**a**) NDWI extraction result, (**b**) MNDWI extraction result, (**c**) EWI extraction result, (**d**) AWEIsh extraction result, (**e**) NWI extraction result.

### 3.4.2. Lakes

The extraction results in Figure 6 show that the MNDWI could accurately extract the overall water bodies and, at the same time, extract the lake boundaries, especially the fine water bodies in the Tenggeli Lake area; thus, the MNDWI was used to extract the typical lakes in the study area.

### 3.4.3. Reservoirs

The main reservoirs in the study area are Mengjiawan Reservoir and Changliushui Reservoir. Mengjiawan Reservoir is a rocky, hilly area with a slightly undulating surface and the extraction of this water body can be easily affected by the shadows caused by the mountains on either side of the reservoir. The Changliu River Reservoir is located upstream of Mengjiawan Reservoir, and compared to Mengjiawan Reservoir, the two sides of the Changliu River Reservoir are narrower valleys, which is unfavourable for water body extraction; thus, the applicability of each of the five water body indices was verified separately to study the reservoir area extraction process, as shown in Figure 8. The black boundary in the figure below represents the manually depicted water body boundaries based on Google Earth satellite imagery taken during the same period: the figure shows that the EWI and AWEIsh achieve the best extraction effects, but when comparing the manually depicted reservoir boundaries, the EWI extraction results clearly exceed the manually depicted reservoir boundaries, while the AWEIsh can better fit these water boundaries. An important reason is that AWEIsh has a better ability to identify mountain shadows on both sides of the river, which is the main reason for the better extraction effect. Thus, reservoir information is ultimately extracted in the study area based on the AWEIsh.

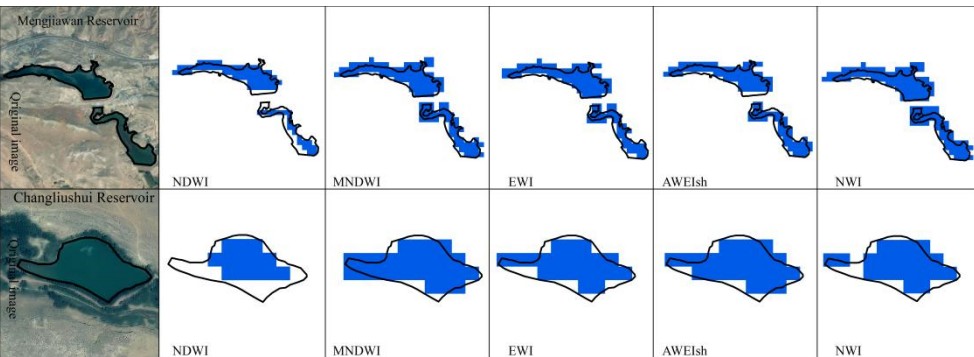

**Figure 8.** Extraction effect of reservoir boundary.

### 3.4.4. Yellow River

As seen from Table 5, the water index with the highest extraction accuracy in the Yellow River is NDWI, with an average accuracy of 90.97% and an average Kappa coefficient of 0.84. In the past ten years, it was the index with the best extraction effect among the five water indices. Similarly, for NDWI, MNDWI, EWI, AWEIsh and NWI, the results are shown in Figure 9. It can be seen that NDWI is the most complete extraction of water bodies, while other indicators all show a certain degree of misclassification. In particular, AWEIsh is the most severe, misdiagnosing large areas of shadow on either side as bodies of water. At the same time, NDWI with the help of NIR bands makes water bodies more visible. The Yellow River has a wide water surface, and its water information is prominent. NDWI was superior to other indices in extracting large water bodies. Combined with the table, the NDWI was used to extract the Yellow River surface water area in a long time series.

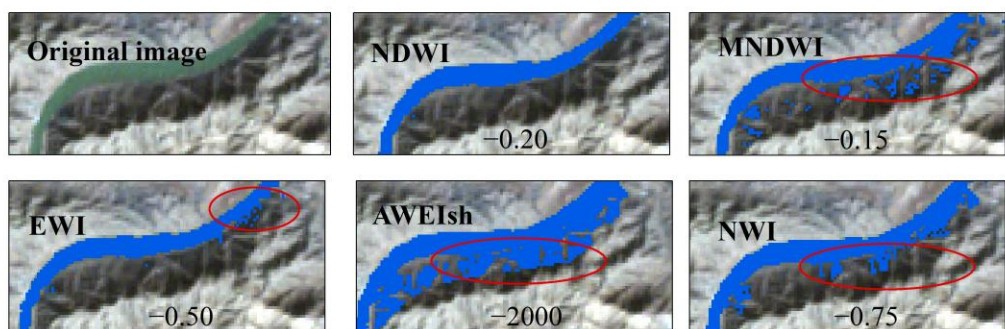

**Figure 9.** Results of extracting the Yellow River using the water body index, the numbers in the figure are thresholds. (The red circle is the area where the water was wrongly extracted).

**Table 5.** Extraction accuracy of different types of water bodies.

| Date | MNDWI (Lake) | | AWEIsh (Reservoir) | | NDWI (Yellow River) | |
|---|---|---|---|---|---|---|
| | OA | Kappa | OA | Kappa | OA | Kappa |
| 2012/3/31 | 87.66% | 0.81 | 92.27% | 0.82 | 97.29% | 0.83 |
| 2013/4/3 | 92.51% | 0.89 | 86.54% | 0.81 | 90.81% | 0.83 |
| 2014/3/21 | 94.63% | 0.87 | 87.19% | 0.87 | 92.08% | 0.83 |
| 2015/3/24 | 86.44% | 0.85 | 89.63% | 0.84 | 84.33% | 0.79 |
| 2016/3/26 | 87.57% | 0.90 | 94.40% | 0.90 | 92.66% | 0.87 |
| 2017/4/22 | 92.65% | 0.91 | 87.74% | 0.86 | 90.12% | 0.84 |
| 2018/3/8 | 95.12% | 0.93 | 94.65% | 0.89 | 82.18% | 0.82 |
| 2019/3/3 | 87.39% | 0.86 | 91.11% | 0.83 | 89.25% | 0.81 |
| 2020/3/21 | 95.38% | 0.92 | 89.20% | 0.84 | 96.49% | 0.92 |
| 2021/3/8 | 97.41% | 0.82 | 95.15% | 0.85 | 94.46% | 0.83 |
| Average Value | 91.68% | 0.88 | 90.79% | 0.85 | 90.97% | 0.84 |

### 3.4.5. Optimal Threshold and Accuracy Verification Results

Through the above experiments, it has been confirmed that the MNDWI, AWEIsh and NDWI are used for water feature extraction for the TSWA (including lakes), reservoirs and Yellow River in the study area. Through the optimal threshold experiment, the optimal threshold of each period of data was determined and the confusion matrix was calculated to obtain the extraction accuracy. The summary is shown in Table 6. The results showed that the average accuracies of the MNDWI, NDWI and AWEIsh were 90.38% (kappa = 0.85), 90.33 (kappa = 0.78) and 90.36% (kappa = 0.82), respectively. The accuracy of the three water indices met the requirements and the extraction of surface water in the time series was completed according to the threshold value in the table.

**Table 6.** The optimal threshold and precision of time series.

| Date | MNDWI (TSWA) | | | NDWI (Yellow River) | | | AWEIsh (Reservoir) | | |
|------|-----------|------|-------|-----------|------|-------|-----------|------|-------|
| | Threshold | OA | Kappa | Threshold | OA | Kappa | Threshold | OA | Kappa |
| 1992/4/1 | −0.15 | 89.54% | 0.80 | −0.15 | 87.96% | 0.74 | −2000 | 90.53% | 0.65 |
| 1993/3/19 | −0.15 | 90.09% | 0.83 | −0.15 | 86.68% | 0.73 | −2000 | 84.75% | 0.84 |
| 1994/3/6 | −0.15 | 93.36% | 0.91 | −0.15 | 95.01% | 0.85 | −2000 | 91.41% | 0.82 |
| 1995/3/25 | −0.2 | 93.12% | 0.87 | −0.15 | 90.27% | 0.81 | −2000 | 90.57% | 0.82 |
| 1997/3/30 | −0.15 | 89.35% | 0.87 | −0.15 | 93.47% | 0.77 | −2000 | 86.80% | 0.79 |
| 1998/4/2 | −0.15 | 94.75% | 0.91 | −0.15 | 93.85% | 0.79 | −2000 | 92.82% | 0.90 |
| 1999/4/5 | −0.15 | 90.23% | 0.88 | −0.15 | 92.45% | 0.73 | −2000 | 91.62% | 0.82 |
| 2000/3/30 | −0.15 | 92.41% | 0.82 | −0.15 | 89.12% | 0.58 | −2000 | 95.13% | 0.84 |
| 2001/3/17 | −0.2 | 86.93% | 0.81 | −0.15 | 97.51% | 0.93 | −2000 | 91.89% | 0.86 |
| 2002/3/28 | −0.15 | 89.10% | 0.81 | −0.15 | 90.16% | 0.68 | −2000 | 86.14% | 0.84 |
| 2003/3/23 | −0.15 | 86.14% | 0.84 | −0.2 | 89.91% | 0.68 | −2000 | 95.64% | 0.88 |
| 2004/3/25 | −0.15 | 90.60% | 0.87 | −0.15 | 92.40% | 0.76 | −2000 | 90.77% | 0.88 |
| 2005/3/4 | −0.15 | 85.13% | 0.83 | −0.15 | 94.77% | 0.85 | −2000 | 91.77% | 0.88 |
| 2006/3/15 | −0.15 | 92.14% | 0.82 | −0.15 | 92.78% | 0.78 | −2000 | 90.39% | 0.89 |
| 2007/5/13 | −0.15 | 87.60% | 0.86 | −0.2 | 87.17% | 0.62 | −1500 | 82.33% | 0.78 |
| 2008/4/13 | −0.15 | 84.33% | 0.81 | −0.2 | 80.97% | 0.62 | −1500 | 89.72% | 0.8 |
| 2009/3/15 | −0.15 | 90.13% | 0.81 | −0.15 | 86.57% | 0.73 | −2000 | 90.25% | 0.81 |
| 2010/4/3 | −0.15 | 90.46% | 0.81 | −0.15 | 93.63% | 0.87 | −2000 | 89.94% | 0.88 |
| 2011/3/29 | −0.15 | 91.22% | 0.88 | −0.15 | 89.41% | 0.79 | −2000 | 90.17% | 0.88 |
| 2012/3/31 | −0.15 | 90.42% | 0.84 | −0.15 | 97.29% | 0.83 | −2000 | 92.27% | 0.82 |
| 2013/4/3 | −0.1 | 92.15% | 0.87 | −0.15 | 90.81% | 0.83 | −2000 | 86.54% | 0.81 |
| 2014/3/21 | −0.15 | 89.34% | 0.83 | −0.15 | 92.08% | 0.83 | −2000 | 87.19% | 0.87 |
| 2015/3/24 | −0.1 | 93.68% | 0.89 | −0.15 | 84.33% | 0.79 | −2000 | 89.63% | 0.84 |
| 2016/3/26 | −0.1 | 89.56% | 0.87 | −0.15 | 92.66% | 0.87 | −2000 | 94.40% | 0.9 |
| 2017/4/22 | 0 | 89.30% | 0.76 | −0.15 | 90.12% | 0.84 | −2000 | 87.74% | 0.86 |
| 2018/3/8 | −0.1 | 89.36% | 0.85 | −0.15 | 82.18% | 0.82 | −2000 | 90.65% | 0.89 |
| 2019/3/3 | −0.15 | 94.23% | 0.90 | −0.2 | 89.25% | 0.81 | −2000 | 91.11% | 0.83 |
| 2020/3/21 | −0.1 | 93.75% | 0.88 | −0.2 | 96.49% | 0.92 | −2000 | 89.20% | 0.84 |
| 2021/1/11 | −0.1 | 95.37% | 0.85 | −0.2 | 90.79% | 0.82 | −1500 | 90.28% | 0.79 |
| 2021/2/20 | −0.1 | 85.41% | 0.85 | −0.2 | 93.63% | 0.87 | −1500 | 89.52% | 0.8 |
| 2021/3/8 | −0.1 | 91.48% | 0.85 | −0.15 | 90.97% | 0.84 | −1500 | 90.79% | 0.85 |
| 2021/5/11 | −0.15 | 92.34% | 0.88 | −0.2 | 90.56% | 0.81 | −1500 | 93.90% | 0.81 |
| 2021/6/4 | −0.15 | 91.56% | 0.90 | −0.2 | 92.33% | 0.85 | −2000 | 94.02% | 0.82 |
| 2021/7/22 | −0.15 | 84.90% | 0.87 | −0.2 | 83.42% | 0.67 | −2000 | 90.66% | 0.7 |
| 2021/8/7 | −0.15 | 91.20% | 0.79 | −0.2 | 80.12% | 0.61 | −2000 | 90.54% | 0.69 |

**Table 6.** *Cont.*

| Date | MNDWI (TSWA) | | | NDWI (Yellow River) | | | AWEIsh (Reservoir) | | |
|---|---|---|---|---|---|---|---|---|---|
| | Threshold | OA | Kappa | Threshold | OA | Kappa | Threshold | OA | Kappa |
| 2021/9/8 | −0.15 | 90.23% | 0.91 | −0.2 | 89.79% | 0.67 | −2000 | 90.78% | 0.7 |
| 2021/11/3 | −0.15 | 93.12% | 0.81 | −0.2 | 89.66% | 0.71 | −2000 | 90.41% | 0.69 |
| 2021/12/13 | −0.15 | 90.23% | 0.89 | −0.15 | 91.91% | 0.76 | −2000 | 91.53% | 0.74 |

*3.5. Annual Variation Pattern of the TSWA*

To illustrate the intra-annual variability characteristics of the TSWA, the year with the most complete data was chosen: 2021, and the results are shown in Figure 10. The optimal threshold values are determined by threshold experiments, as shown in the Table 6.

It can be seen that there is a trend of increasing and then decreasing TSWA during the year. Although the data for April are missing, it is easy to tell from the folding trend that February, March and April of this year are the most abundant periods of the TSWA and that surface water resources are the most abundant phase of the year. It reaches its lowest value in July, August and September of the year, which is also due to the strong evaporation in the study area during that period. It can be deduced that without artificial irrigation, the situation will worsen, with water resources continuing to deteriorate and even drying up during this period. In autumn, however, evaporation gradually decreases as the temperature drops, and with artificial recharge, the water area gradually increases again at this time. It is thus easy to see that artificial recharge of the study area is important during the summer months of each year. At the same time, our choice of data near March each year for the extraction of interannual variability features of the water bodies is accurate.

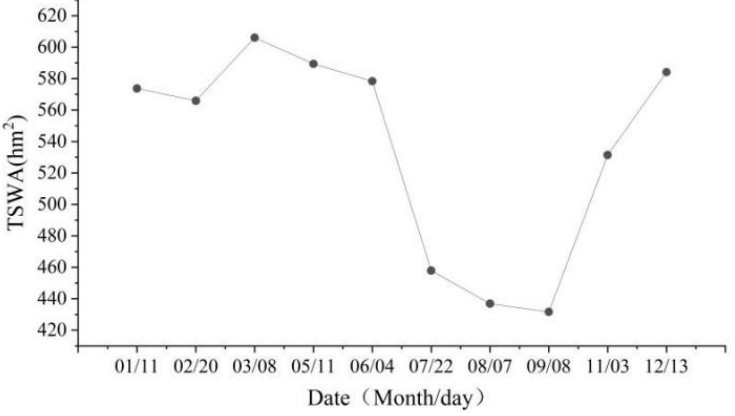

**Figure 10.** Month-by-month water body extraction results for 2021.

*3.6. Interannual Variation Pattern of the Water Body Area*

Surface water area was extracted from TSWA, lakes and reservoirs in the study area between 1992 and 2021 to obtain a dynamic change curve of the surface area of water bodies over a thirty-year period. Based on Equation (6), a slope value > 0 indicates that the water body area has increased, while a slope value < 0 indicates that the water body area has decreased. The TSWA trend change calculation results obtained for the study area and the water body area change trends corresponding to each lake and reservoir are shown in Table 5; the results are all greater than 0. Based on the Mann–Kendall significance test results, all extractions passed the 99% significance test except those for Mengjiawan Reservoir, which did not pass the significance test, and Machang Lake, which passed the significance test at the 90% confidence level (Table 7).

**Table 7.** Results of Trend Slope Analysis.

| Water Body | Slope | Man-Kendall |
|---|---|---|
| TSWA | 0.2293 | 5.34 |
| Tenggeli Lake | 0.0946 | 5.73 |
| Gaodun Lake | 0.0155 | 3.73 |
| Machang Lake | 0.0059 | 1.59 |
| Xiaohu Lake | 0.0516 | 4.87 |
| Qiandao Lake | 0.0057 | 4.43 |
| Mengjiawan Reservoir | 0.0016 | −0.23 |
| Changliushui Reservoir | 0.0003 | 3.32 |

### 3.6.1. Interannual TSWA Variation

The TSWA analysis for the study area shows the following. For the 30-year TSWA for the period 1992–2021, a slope value of 0.2293 was calculated and passed the significance test. This reveals an increasing trend in the TSWA, as shown in Figure 11a. The TSWA in 1992 was 223.78 hm$^2$ and in 2021, the TSWA was 606.06 hm$^2$, representing an increase of 382.28 hm$^2$ over 30 years and a combined multiyear average TSWA of 447.98 hm$^2$. The main time point of area increase was in 2004 at 471.97 hm$^2$, with a subsequent increase to a maximum of 752.71 hm$^2$ in 2015. The area then changed less and gradually stabilised at approximately 650 hm$^2$. The TSWA in 2004 showed a dramatic increase, which is to be explained in the context of the reality of the study area to better explain the changes that occurred during that period. One of the main reasons for this is the expansion of the area of the lakes during this period, with Tenggeli Lake showing the largest expansion, followed by Gaodun Lake and Machang Lake. Second, aquaculture in the study area gradually increased in size with socioeconomic development, and the large number of new fishponds built during this period is also an important reason for the increase in surface water area. At the same time, the citation of water from the Yellow River and the influence of artificial recharge also led to a further expansion of the water surface area.

### 3.6.2. Interannual Variations in the Areas of Major Lakes and Reservoirs

In this paper, we analysed the interannual variation characteristics of the water body areas of five major lakes and two reservoirs in the study area and the calculation results of each typical lake and reservoir are shown in Figure 11. Overall, the area of each lake and reservoir increased and there was no obvious reduction phenomenon. The water body areas increased the most for Tenggeli Lake, Xiaohu Lake and Qiandao Lake. Combining these results with those in Table 5, the areas of Tenggeli Lake and Xiaohu Lake increased the most, with slope values of 0.0946 and 0.0516, respectively. This paper analyses the interannual variation characteristics of the water area of five major lakes and two reservoirs in the study area during the period 1992–2021, and the calculated results for each typical lake and reservoir are shown in Table 2 and Figure 10.

Tenggeli Lake is the largest artificial lake in the study area, increasing by 257.27 hm$^2$ between 1992 and 2021, with a multiyear average area of 184.09 hm$^2$; the main increases in area since 1992 occurred at three time points, 1997, 2002 and 2010, from 31.55 hm$^2$ to 73.60 hm$^2$ and 111.76 hm$^2$. From 1992 to 2009, the Tenggeli Lake area was an artificial fishpond during this period. The increase in area in 1997 and 2002 was mainly due to the expansion of fish farming by local residents and thus the expansion of fishponds, which led to an increase in the area of water bodies in the area. The fishponds were further deepened and enlarged after 2009, resulting in the gradual formation of Tenggeli Lake. After 2010, the area of the water body remained stable, with an average area of 292.27 hm$^2$ over the last five years.

Xiaohu Lake and Qiandao Lake are typical seminatural lakes in the study area; these lakes were formed through artificial transformation while maintaining the original lake basin. The artificial transformation of Xiaohu Lake started mainly in 2003, and its area increased by nearly 15 hm$^2$ relative to the same period in 2002; then, the lake experienced a larger increase in area in 2005, growing by 66.82 hm$^2$ compared to the previous year's area (21.74 hm$^2$). The increases in the areas of Qiandao Lake and Xiaohu Lake occurred at the same stage, with large increases observed in 2005 compared to the previous year (the 2004 area of Qiandao Lake was 1.40 hm$^2$, indicating an increase of 17.48 hm$^2$). In general, the area of Xiaohu Lake has increased significantly since 2002, increasing by 15.08 hm$^2$ by 2003 for an increase of approximately 250% and then increasing to 162.91 hm$^2$ by 2021. In contrast, the main phase of the increase in the area of Qiandao Lake occurred in 2005, and since then, the lake has essentially not increased significantly again, in stark contrast to Xiaohu Lake. Considering this difference, we conclude that in addition to both lakes being artificially recharged, the continued increase in the area of Xiaohu Lake is also explained by the lateral infiltration of residual irrigation water from the surrounding poplar forests, which resulted in the rapid expansion of the lake in 2003 and the maintenance of this expanded lake area thereafter.

Gaodun Lake and Machang Lake are natural lakes in the study area that have been watered since 1992, for which complete 30-year data were extracted. The multiyear averages for these two lakes are 30.81 hm$^2$ and 11.78 hm$^2$, respectively, with larger averages of 36.57 hm$^2$ and 14.89 hm$^2$ for the last five years. Both lakes have increased in size to some extent, and their water areas have been effectively maintained in recent years. The maintenance of the water area of Machang Lake has mainly been due to the local residents developing this lake as a fishpond for water storage and aquaculture. Gaodun Lake, on the other hand, benefits from local ecotourism and maintains its water area through artificial water injections. In both cases, a better water surface is maintained, and the water body areas can hardly be sustained without artificial intervention together with the natural state of lateral seepage, the transpiration of aquatic plants and evaporation from the land surface. In addition, the southern sections of Machang Lake and Gaodun Lake are adjacent to Tenggeli Lake and are thus affected by the lateral leakage of the high water level of Tenggeli Lake, allowing them to maintain highly stable water body areas under the influence of the high water level of Tenggeli Lake in recent years.

Mengjiawan Reservoir and Changliushui Reservoir are the main reservoirs in the study area. Mengjiawan Reservoir has been monitored, as it has not been dry, with a multiyear average area of 3.31 hm$^2$ and a recent five-year average area of 3.38 hm$^2$. Changliushui Reservoir has been dry for many years, with a multiyear average area of 0.55 hm$^2$ and a recent five-year average area of 1.09 hm$^2$. The recent five-year average areas of these two reservoirs are higher than their multiyear average areas, further indicating the relative stability of these reservoir areas in recent years. Regarding the interannual variation characteristics of these reservoirs, both are located in the western Changliushui area within the study area, a river region formed by outcropping springs, and the reservoirs have limited water storage sources and are responsible for holding irrigation water and domestic drinking water used by people living in Mengjiawan and Changliushui villages, thus resulting in their large fluctuations over the years with very little recharge or production and their application for domestic water use.

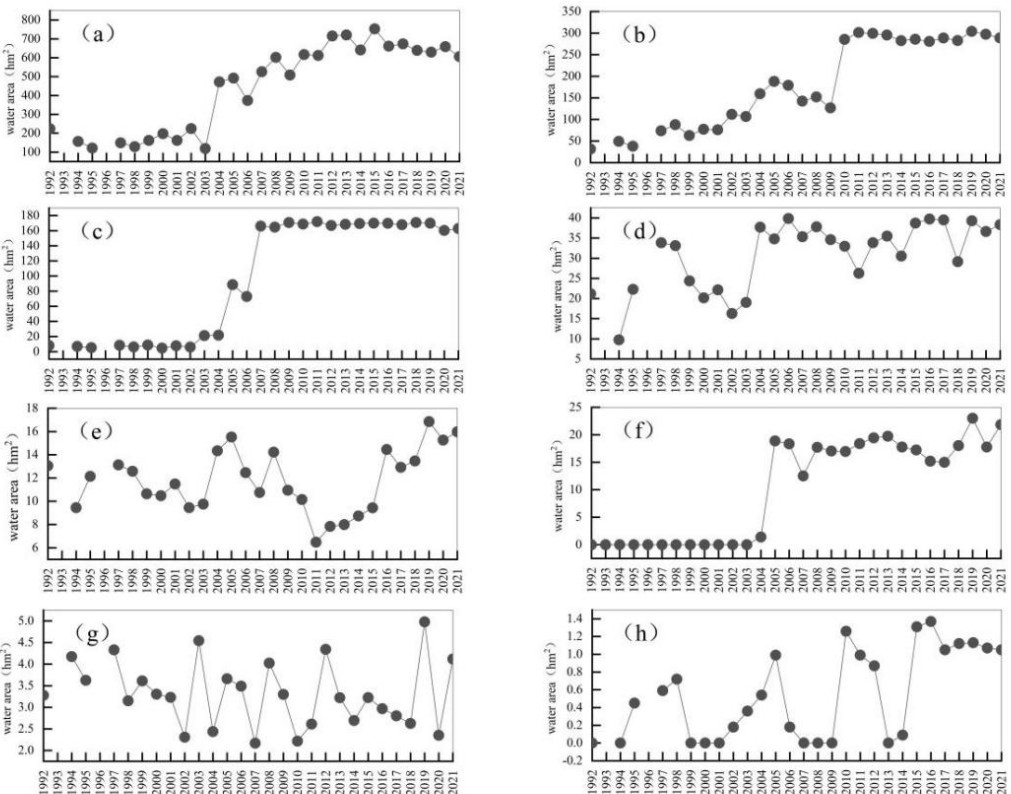

**Figure 11.** Interannual variation in water body area: (**a**) TWSA, (**b**) Tenggeli Lake, (**c**) Xiaohu Lake, (**d**) Gaodun Lake, (**e**) Machang Lake, (**f**) Qiandao Lake, (**g**) Mengjiawan Reservoir, (**h**) Changliushui Reservoir.

### 3.6.3. Characteristics of Interannual Variation in Surface Water Area of the Yellow River

The multiyear average surface water area of the Yellow River flowing through Zhongwei city from 1992 to 2021 was 1732.75 hm$^2$, and the surface water area in 2021 was 1746.04 hm$^2$. In general, the 30-year average surface water area was 1732.75 hm$^2$, and the area shows an increase. This is generally consistent with the results of Li's surface water area monitoring for the whole Yellow River basin [46]. At the same time, there is a minimum value in 2003, the year in which Li's study noted a significant reduction in surface water area in the Yellow River source area and we speculate that it is this that has led to the phenomenon of a minimum value in this paper. The area of water bodies has been relatively stable over the last five years, with an average area of 1841.39 hm$^2$ (Figure 12).

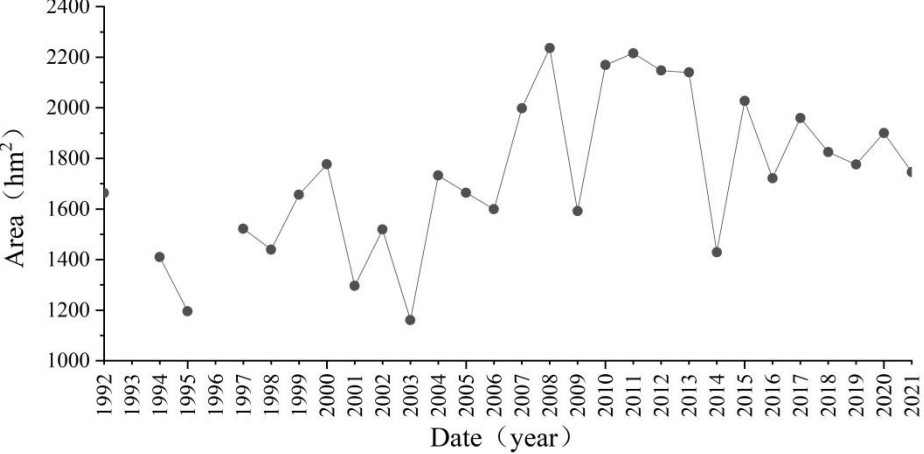

**Figure 12.** Surface water area of the Yellow River.

### 3.7. Correlation Analysis

According to Equation (10), correlations were calculated between the TSWA and prior year TSWA and surface water area of the Yellow River for each year from 1992–2021. The results were calculated as 0.89 and 0.71 respectively (Figure 13)

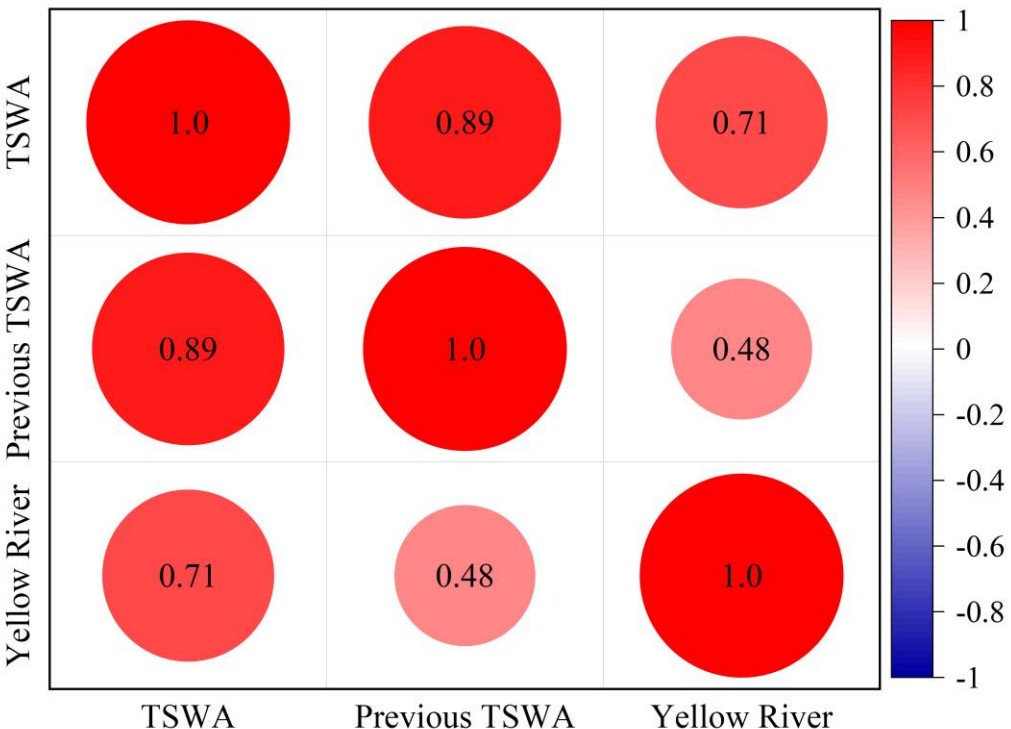

**Figure 13.** Correlation coefficient graph.

The correlation coefficient derived between the current and previous-year TSWAs reached a maximum value of 0.89 and showed significance at the 0.01 level, mainly because the previous-year TSWA played a crucial role in the maintenance of the water body area from that time forwards, allowing the overall storage size in the previous year to have continuity; this continuity influenced the water body area in the second year and there were very few natural recharge sources in the study area, causing the water body areas to be dependent only on artificial recharge. This condition caused the previous-year TSWA to play a crucial role with respect to the total water body area.

The correlation between the surface water area of the Yellow River and TSWA for the same period is second only to that between TSWA and TSWA of the previous year, with the former showing a correlation coefficient of 0.71, showing significance at the 0.01 level. This is partly because the study area is adjacent to the Yellow River, which recharges most of the water bodies in the study area, especially the typical lakes in the study area. On the other hand, it is also due to the combined effect of the Yellow River and the major water bodies in the study area in terms of topographic conditions and permeability, which has produced contemporaneous groundwater volume interactions that have had a positive effect on maintaining the water body area in the study area. The subsequent use of the Yellow River water source to recharge the water resources of the study area in a more rational way deserves deeper reflection with respect to the water scarcity dynamics of the study area, especially during the summer months when evaporation is strong due to rising temperatures and great instability in the TSWA.

## 4. Conclusions

It is still unclear how to address the problem caused by the characteristics of long-term dynamic changes in surface water area and the influence mechanism of TSWA trend changes in the Sha Botou National Nature Reserve of Ningxia. In this paper, we produce a time series dataset of TSWA and each typical water body in the study area at 30 m resolution from 1992 to 2021 based on long time series Landsat data and water indices and reveal the characteristics of interannual changes affecting local water bodies and the associated impact mechanisms. This work complements the existing methods for extracting information on surface water resources in the study area and is also of great practical importance for the scientific management of water resources in the study area.

The main conclusions of this study are as follows:

(1) The validation results of the selected water body samples from Google Earth satellite imagery show that the three water body remote sensing indices NDWI, MNDWI and AWEIsh considered in this paper have high confidence in the extraction of water bodies in the study area, with average overall accuracies of 90.38%, 90.33% and 90.36%, respectively, and that the extraction results can adequately reflect the interannual dynamic trajectories of various surface water bodies. Compared with the support vector machine classification method, the water index method is more reliable after a strict threshold selection.

(2) The TSWA in the study area showed an overall increasing trend between 1992 and 2021, from 223.78 hm$^2$ in 1992 to 606.06 hm$^2$ in 2021, with a multiyear average TSWA of 447.98 hm$^2$. The main increase in area was caused by Tenggeli Lake, which increased by 257.27 hm$^2$ by 2021.

(3) Pearson correlation coefficients were used to evaluate the correlation between each consideration and TSWA, and the correlation between water body extraction results and previous year TSWA and surface water area of the Yellow River over a 30-year period was analysed. The results showed that the correlations between the previous year TSWA and the TSWA were greater than those between the surface water area of the Yellow River and the TSWA. The influence of the previous year TSWA on the TSWA was the largest at 0.89, followed by the surface water area of the Yellow River at 0.71 for the same period. This also suggests that current water conservation in the study area should focus more on anthropogenic impacts.

**Author Contributions:** Conceptualisation, H.P. and X.W.; methodology, H.P.; software, H.P. and R.H.; validation, H.P., X.W., Z.B. and G.S.; formal analysis, H.P.; investigation, H.P., X.W. and W.Y.; resources, Z.B.; data curation, H.P.; writing—original draft preparation, H.P.; writing—review and editing, H.P.; visualisation, H.P.; supervision, H.P.; project administration, H.P.; funding acquisition, Z.B. All authors have read and agreed to the published version of the manuscript.

**Funding:** This research was funded by National Natural Science Foundation of China (31400619) and the National Nature Reserve Science and Technology Research Project (YL-2021122101).

**Data Availability Statement:** The authors thank editors and anonymous reviewers.

**Conflicts of Interest:** The authors declare no conflict of interest.

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
