# Peer review of "Multiwater Index Synergistic Monitoring of Typical Wetland Water Bodies in the Arid Regions of West-Central Ningxia over 30 Years"

_water, doi:10.3390/w15010020_

Round 1
Reviewer 1 Report (Previous Reviewer 1)
Thank you for submitting your revised paper to Water. I read carefully manuscript number: water-2099415, the manuscript entitled: "Multiwater index synergistic monitoring of typical wetland water bodies in the arid regions of west-central Ningxia over 30 years ". In my point of view, the result of this kind of research could be interesting and useful for many applications specifically for the spatial inundation and risk mapping of multi-index analysis. All previous comments were applied. The authors applied all comments point by point and I confirm their revision. The added information is important and useful and led to improving the manuscript. I accept the revised manuscript in this present form. I concur; the final decision is accepted for publication.
Author Response
Dear Professor:
I would like to express my appreciation.
Reviewer 2 Report (Previous Reviewer 2)
All my concerns are well anwsered.
Author Response
Dear Professor:
I should like to express my appreciation.
Reviewer 3 Report (New Reviewer)
The article is entitled “Multiwater index synergistic monitoring of typical wetland water bodies in the arid regions of west-central Ningxia over 30 years”.
The paper compares indicators to characterize surface water using Google Earth satellite imagery and shows the dynamics of changes in the total surface water area over a 30-year period.
The characteristics of the Ningxia Hui Autonomous Region in China, where the Shapotou National Nature Reserve is located, are given in the work. The following is about the Yellow River. In order to understand in what connection the river is mentioned, in my opinion, it should be noted that the Yellow River flows through the Shapotous nature reserve.
Please do not be offended by me, but I did not quite understand some phrases (maybe these are translation errors into English):
What do you mean by “to extract the TSWA”? Extraction is from the field of chemistry. Maybe - "to calculate/determine the TSWA"?
Line 440: “The Yellow River has a wide water surface, and its water information is prominent.” What do you mean by “its water information is prominent”?
TSWA is the total surface water area. It is not always clear in the text which TSWAs you are comparing. For example, for lines 643-644: “…the previous year TSWA on the TSWA was…”
Author Response
Please see the attachment.

This manuscript is a resubmission of an earlier submission. The following is a list of the peer review reports and author responses from that submission.
Round 1
Reviewer 1 Report
Reviewer comments
Thank you for submitting your paper to Journal of Water. I read carefully manuscript number: water-1957043, the manuscript entitled: "Multi-water index synergistic monitoring of typical wetland water bodies in the arid regions of west-central Ningxia over 30 years". The current research, by using the common water-body remote sensing indices (NDWI, MNDWI, EWI, AWEI and NWI), extract the multiyear water-body area in Zhongwei. In my point of view, the result of this kind of research could be interesting and useful for many applications specifically for the spatial-temporal water detection, and inundation mapping. The English language is moderate. Some sections of paper require major revisions before any further. I attached my reviewer supplementary comments in the below and manuscript PDF file.
1- Abstract
1-1- The abstract section need to complete with more information. The abstract should be improved.
1-2-The concrete finding of this research need to be added to the abstract section.
2- Introduction
2-1- The literature review is too general and thus can’t indicate any novelty of the current study. It is better that explain more about the novelty of manuscript in introduction section. The manuscript has not quite innovative. Please explain about its novelty.
2-2- Research organization not provided.
3- Research Methodology section were provided in poor way. So needs improvements:
3-1- Methodologies and Threshold selection in the manuscript should describe clearly.
4 -"Results and Discussion" were provided in poor way.
4-1-Results of this study need to be compared with previous research works. Authors are emphatically recommended to provide a new section for this purpose.
4-2- Authors are recommended to use more statistical measures and classification analysis by data-mining techniques (e.g., support vector machine, random forest, model tree, and decision tree).
The following reference can be useful:
Farhadi H, Esmaeily A, Najafzadeh M. Flood monitoring by integration of Remote Sensing technique and Multi-Criteria Decision Making method. Computers & Geosciences. 2022 Mar 1;160:105045.
5- Conclusion section need rewriting.
6- The punctuation marks here and the entire text of the article should be corrected.
Some of them are shown below:
Line 159 - from 1992-2021(Table 1),
Line 171 - Figure 2. Repair of missing strip problems(a:Before restoration;b:After the repair is completed)
Line 283- Figure 5. Extraction of images for effects(a:NDWI;b:MNDWI;c:EWI;d:AWEIsh;e:NWI),
Line 181 - water bodies(In order to…,
Line 310
Line 346,…..

Author Response
请看附件

Reviewer 2 Report
The author investigates water bodies in the arid regions by multiwater indices, it is meaningful and help us understand more about wetland water bodies in the arid regions. Specific comments are as follows:
1. AWEI in abstract should be AWEIsh.
2. Legend should be added in Figure 2 and Figure 9. And font-size in Figure 4 are very small and unclear.
3. In Section 2.1, why author show temperature and precipitation in two period, i.e. 1959-2021 and 2012-2021, and data source for evaporation.
4. Section 3.4.1, phenomenon of the increased TSWA is very important and I think the reason is not clear in this paper. Water area is sharp increasing around 2003 (Figure 9a,c,f), and 2009 (Figure 9b), are they all artificial water injection recharge in one year and then the recharge maintains stable to balance evaporation?
5. The author only used a specific month, there is no seasonal variation can be found here, so is it any seasonal variation of the TSWA?
6. The author states that the TSWA shows weak correlate to the precipitation and moderate with temperature. I think this content is not appropriate in this study, due to water body is intervened by human injected recharge. And relationships between the TSWA and meteorological factors are all meaningless.
Reviewer 3 Report
Overall, I find the paper to be hard to follow, given the poor structure, questionable/ unclear approaches, and lack of discussion about the sources of error.
Please find my comments / suggestions bellow:
17: Check if the numeric notation found in the abstract body is according to the journal layout requirements.
52: Arguable sentence.. other studies may suggest otherwise. It depends on many factors (I suggest reading the following literature):
· Zhou Y, Dong J, Xiao X, et al (2017) Open surface water mapping algorithms: A comparison of water-related spectral indices and sensors. Water (Switzerland). doi: 10.3390/w9040256
· Rokni K, Ahmad A, Selamat A, Hazini S (2014) Water feature extraction and change detection using multitemporal landsat imagery. Remote Sens 6:4173–4189. doi: 10.3390/rs6054173
98: The mentioned spatial resolutions only refer to the multispectral bands from visible to shortwave infrared (not the same for panchromatic or thermal bands)
101: Landsat 9 is already operational
111: Justify the choice of these indices
113: The objectives should be clarified in terms of what reference data is being use (and how it is determined), and how are the threshold for these indices are determined and later incorporated in the multi-year analysis.
Figure 1: What does the image on the right represent? Satellite composition? Aerial photo? This information should be given (also the date when acquired). The authors could also include a delimitation/path line for the Yellow River and the Zhongwei city as toponymy.
127: It is unclear what is the study area really is. Does it correspond to the limits of the Shapotou National Nature Reserve, or just a section of it? What are the sections of the Yellow River which are part of the study area? Are there any other major rivers? If so, where are they located?
158: what are phase-one images? Is it related with Landsat Level 1 or Collection 1 products?
Why choosing the images in this period? Does it correspond to the period in the year with minimum levels? Or reduced cloud cover? This issue is very important to be justified, especially because of the monsoon effects.
163: Did the authors performed the radiometric and atmospheric correction, or used Level 2 products? If so, further details should be given.
165: Confusing date notation (please follow the journal template recommendations)
169: Would not call it improved data, but instead filled data gaps through means of interpolation(?) Why not use the LS8 for those periods?
Figure 2: the legend should include the details of the scene being used (i.e., Sensor, product level, date, coordinates/scale, etc.).
189: This whole Chapter is very confusing. It is not clear how each section is connected to the others. Perhaps, the paper could benefit with a workflow diagram.
191: The choice of these indices (especially after introducing many others) should be justified
194: The lakes names do not seem to correspond entirely with Figure 3.
Why did you choose a LS8 image from August 2019? It doesn’t seem to be consistent with the period stated in Section 2.2.1, neither those of 2.2.2
201: if the ρ symbol is used it is best to refer it as surface reflectance
203: There should be a section explaining why and how the optimal thresholds were selected (with references). Be aware that these thresholds are both sensor and scene dependent. These thresholds can also change depending on water properties (e.g., concentrations of phytoplankton and sediments, depth, or substratum type - suggest reading the following literature):
· Oliveira ER, Disperati L, Cenci L, et al (2019) Multi-Index Image Differencing Method (MINDED) for flood extent estimations. Remote Sens 11:1–29. doi: 10.3390/rs11111305
· Fisher A, Flood N, Danaher T (2016) Comparing Landsat water index methods for automated water classification in eastern Australia. Remote Sens Environ 175:167–182. doi: 10.1016/j.rse.2015.12.055
209: What do the authors mean by validation point data? Did they select a certain number of validation points within the study area (if so, how many)? Or did they classified the entire watershed area from the google earth images? Further details about the reference must be given. Is this only in respect to the Landsat 8 image from august 2018?
217: How is thiewater body area time series determined? With the indices? If so, are they calculated for all indices individually? What are sky map images?
249: Again, this whole section is also very unclear. What do these thresholds refer to, i.e., which lake or area is being analysed? How was the reference google earth image surface extracted? Why choosing a Landsat 7 image from 2018.03.03? How does it relate with the previously mentioned Landsat 8 from August 2019? Why is a Landsat 7 image from 2018.03.03 being compared to the Google earth images from March 21, 2019 and July 12, 2018? What is thresholding step considered for calculating the indices?
263: Figure 4 is unreadable at the provided resolution.
Figure 5: What and where does the area in the top left corner corresponds to?
270: Confusing nomenclature: What do the authors mean by Total Water Bodies? Lakes and rivers are also water bodies..
283: Is the Tenggeli Lake area referring to the comment of Figure 5? Again, how were the reference lake limits extracted and were can the reader find the representation of these reference limits?
293: Upper reaches of what?
300: such classification of wet soil / water surface may only be verified with ground level confirmation.
307: The previously calculated optimal thresholds should be highly dependent on sediment concentration (particularly for NIR bands, as the author refers). This means that the correlations with the reference water body limits could be dependent on to such concentration variations.
312: Do you the authors have a ground level validation of the Yellow River limits? With the provided images, it is not possible to conclude that the NDWI achieves the best representation (perhaps including the corresponding google earth image for this section could help).
318: Which small lake? This section needs further clarification.
323: Clarify the meaning of inversion accuracy/inverting water body (if necessary, include references).
328: Please clarify which period is being analyzed in this section? This subtitle needs to be formatted according to the layout.
347: Again layout issues. This whole section is very hard to follow. Figure 9 has no legend about what each graph corresponds to.
414: Please clarify if the frequency of these correlations (i.e., once per year?) and what to the results in Figure 10 correspond to (average?)
464: It should be highlighted that the indices were compared to manual classifications of Google Earth images.
472: The authors must clarify how this multiyear analysis was performed. Also, in the conclusion, the period under analysis should also be highlighted.
Round 2
Reviewer 1 Report
Thank you for submitting your revised paper to Water. I read carefully manuscript number: water-1957043, the manuscript entitled: " Multiwater index synergistic monitoring of typical wetland water bodies in the arid regions of west-central Ningxia over 30 years Prefecture ". In my point of view, the result of this kind of research could be interesting and useful for many applications specifically for the spatial inundation and risk mapping of multi-index analysis. All previous comments were applied. The authors applied all comments point by point and I confirm their revision. The added information is important and useful and led to improving the manuscript. I accept the revised manuscript in this present form. I concur; the final decision is accepted for publication.
Reviewer 3 Report
Dear authors,
Despite acknowledging some efforts in providing some additional details about your methodological approach, in my opinion Chapters 2 and 3 continue to be very confusing. I find this second version of the manuscript to be more affected by severe English issues, including many grammar mistakes (e.g., entire paragraphs without any verb conjugation), excessive long sentences and inconsistent use of terminologies. Such issues make it difficult to understand what seems to be an already confusing approach. Most of all, I struggle in understanding how the different sections are connected with each other. I believe, investing in significantly improving the workflow could help to resolve some of this issues. Moreover, I suggest to fully reformulate Chapters 2 and 3 following the same order of the workflow.
Finally, I found the discussion provided with the Results & Discussion chapter to be insufficient and I consider that the paper would likely benefit from including a general discussion chapter/section. Such general discussions should be useful for clarifying the relationships between every section of Methods and Results, as well as highlighting the main strengths and drawbacks of your approach (supported by literature references).
Aditionally, please find the following specific comments/suggestions:
· Another pdf version without track changes (or the word file document) should be provided to facilitate subsequent reviews.
· 78: the nomenclature of NWI is not consistent with line 272 (new/novel water index).
· 112-116: Grammar checking required.
· 117: Landsat 9 has already been launched so this sentence must be corrected.
· The last paragraph of the introduction is very confusing and requires extensive grammar checking. The objectives of the research should be concise but clear (using action verbs in the past tense is often a good strategy). I suggest clearly listing the need for your work and briefly indicating what you have done to address such need.
· Extensive revision of English is also required in chapters 2 and 3 which are still very hard to understand.
· The new paragraph in the beginning of Chapter 2 is hard to follow (should be split into several sentences).
· The workflow needs to be vastly improved. For example, in which step(s) do you perform interpretation of satellite imagery (i.e., Google Earth or any other platform), and in when do you use such data? In which step is the threshold selection procedure implemented and later integrated in the further steps of the workflow? It should be very clear which are the main inputs and outputs of every step and how they are integrated with each other. I suggest creating a flowchart using different shapes (e.g., rectangles for processes, diamonds for decisions, parallelograms for Input/Output data, etc.).
· For me, chapters 2 and 3 are very confusing, with several steps being mixed and/or out of order. I suggest full restructuring following the same order of the workflow.
· In 2.3.2, were these points randomly selected or manually assigned by the authors? This issue is very important to be mentioned and later discussed because it will have a major effect when calculating the accuracies of the indices. If the sampling points are only selected in “easy” areas, the accuracies are expected to be high, whereas if they include transition areas between “water” and “non-water”, such classification will be affected by uncertainties caused by the spatially continuous variation of soil wetness. These uncertainties should affect any method, including the multispectral indices and the visual interpretation of satellite images. In my opinion, the authors should acknowledge the expected user induced errors from the visual interpretation of Google Earth and Satellite Images. Such errors will naturally occur when classifying the sampling points without having ground truth, but especially when extracting the polygons of the water bodies. The paper should also discuss that despite being reasonable to use them as reference to determine accuracy, such datasets should never be expected to have the same accuracy of ground level data.
· 202: The specific sensor used in Figure 2 should be mentioned (e.g., Landsat 8 OLI).
· 219: Again, the details about such corrections should be included in the paper (i.e., which methods and eventually the settings of whatever used software). Also, why not consider instead the Level 2 products, which are already radiometrically and atmospherically corrected?
· 232: Should be Google Earth satellite imagery instead of Google Satellite imagery.
· 281-282: It should be mentioned that the sampling points were classified with visual interpretation of both Google Earth images and Landsat data.
Please use a consistent designation for accuracy verification/ sampling points.
· The choice of different thresholding steps for normalized and non-normalized indices should be justified.
· The paper continues to have an unclear structure. For me it is not clear how the different steps are connected with each other. Also the methods and results seem to appear in different orders.
· In 3.1 it is not clear which area was considered for the optimal threshold experiment (which again should have a consistent designation with 2.3.3). Was it for the sampling points mentioned in 2.3.2? Before including the accuracy graphs, there should be a map, or perhaps even a new section with the details about the location of the sampling points, their classification and the details of the visual interpretation from Google Earth/Landsat (e.g., was it performed by one or several users, strategies for dealing with dubious areas, etc.).
· In Figure 4 c) to e), the legend of the series ‘Overall Accuracy’ does not seem to be correct.
· It is hard to understand how 3.1, 3.2, 3.3 and 3.4 are connected. They all seem to be compared with different reference areas (“sampling points,” “study area”, “extraction area”, “study area”) which is very confusing.
· In 3.2, which is the period being analyzed? What are "visual extraction effects"? How did you obtained the reference polygons for the water bodies? If you obtained such reference data with visual interpretation of Google Earth/Landsat, what is the purpose of section 3.1 and why use sampling points instead? Also, if you have such reference water deliniation polygons, it would be better to calculate accuracies (in a similar way to Table 2). Otherwise the choice of the most suited indices is very subjective. For example, from Figure 5 alone it is not clear how MNDWI performs better in comparison to the other indices.
· In 3.2 the authors could include a discussion about the reasons behind different indices having different responses depending on the water body type.
· For Figure 8 there is no reference to the meaning of the the red polygon. If the authors have determined the limits for the yellow river (with visual interpretation of satellite images) they should be included. AS an alternative, a satellite/google earth RGB composition should be included. Otherwise it is not possible to verify the accuracy of each index and thresholds.
· It is not clear which is the period and reference data used to determine the accuracy of SVM.
· 465: This sentence is not in accordance to the results of Table 2 (for 2010, AWEIsh has the best accuracy).
· Again, there is no indication about the reference data used in 3.5. The manuscript must specify if such data was determined or not by the authors. If so, the data and methods for its determination should be indicated.
· Table 4 has data formatting issues